

# 1 Geomorphological landslide inventory map of the Daunia Apennines,

# 2 southern Italy.

Francesca Ardizzone[1], Francesco Bucci[1], Mauro Cardinali[1]*, Federica Fiorucci[1], Luca Pisano[2], Michele
Santangelo[1], Veronica Zumpano[2]
[1]CNR - IRPI, Via della Madonna Alta 126, 06128, Perugia, Italy
[2]CNR - IRPI, Via Amendola 122, 70126, Bari, Italy
*Correspondence to: Mauro Cardinali (mauro.cardinali@irpi.cnr.it)
**Keywords**
Landslide inventory map, geomorphology, landslides, Daunia, expert mapping, Southern Italy
**Abstract**
Detailed and accurate geomorphological historical landslide inventory maps are an invaluable source of information for many
research topics and applications. Their systematic preparation worldwide has been advised by many researchers as it may
foster our knowledge on landslides, their spatial and temporal distribution, their potential interaction with the built
environment, their contribution to landscape dynamics, their response to climate change in the past. Due to the extreme
variability of the morphological and radiometric elements that can reveal historical landslides, geomorphological historical
landslide inventory maps are produced by expert interpretation, which makes it a time consuming and expensive process,
which often discourages wide area mapping activities. In this paper we present a new geomorphological historical landslide
inventory map for a 1,460 km² area in the Daunia Apennines, the northern-western sector of the Apulia (Puglia) region, in
Southern Italy. The inventory contains 17,437 landslides classified according to relative age, type of movement and estimated
depth. Landslides were mapped according to rigorous and reproducible criteria applied by two teams of expert photo-
interpreters to two sets of stereoscopic aerial photographs taken in 1954/55 and 2003. The dataset consists of a digital archive
publicly available at https://doi.pangaea.de/10.1594/PANGAEA.942427 (Cardinali et al., 2022).

**1 Introduction**

Landslides are widespread natural hazards that occur worldwide posing threat to population, structures and infrastructures,
causing relevant economic damage to society (Donnini et al., 2017; Froude and Petley, 2018; Petley, 2012). Availability of
information on landslides locations, types, and sizes is a key basic information for many activities, from research to emergency
support and land planning. Nevertheless, expert landslide mapping is not an investment priority in many countries and
institutions, hence homogeneous wide scale landslide maps are rare (Guzzetti et al., 2012), although the availability of a

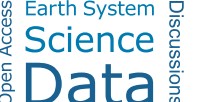

landslide inventory map is fundamental to the production of landslide hazard and risk maps (Thiery et al., 2020). As a result,
several applications spanning from land management to susceptibility modelling that would benefit from more homogeneous
and complete datasets rely on often poor or incomplete/inconsistent data.
Any landslide mapping activity assumes (i) that landslides leave discernible features on the territory, i.e., a "landslide
signature" made by radiometric and morphologic elements (Fiorucci et al., 2018; Guzzetti et al., 2012; Santangelo et al., 2022),
(ii) that, in case of use of remote sensing, the data used (e.g. images) portray them, (iii) that the technique adopted is adequate
to the data used, and (iv) that the operator (or the automatic or semi-automatic classification system) is able to exploit those
features to detect and map landslides. Therefore, any landslide mapping activity is affected by a degree of incompleteness and
inconsistency that stems from these basic assumptions. Landslides can remain unnoticed since (i) features were cancelled by
erosion, other landslides and/or human activity (Malamud et al., 2004), (ii) the type of image or its spectral and spatial
resolution may be inadequate for some types or sizes of landslides or for some locations (e.g. slope orientation), (iii) interpreters
or (semi-)automatic systems may be unable to map some landslides using certain images or techniques (e.g. lack of experience,
poor or incorrect system training).

Landslide inventory maps (LIMs) are the simplest tools to represent landslides distribution in a territory. They usually display
landslide locations preferably as polygons rather than points or lines which should be used only for scale representation issues.
LIMs usually store landslide attributes such as type of movement, estimated time of occurrence, estimated depth, activity at
the date of the observation if available, and other more detailed data depending on availability and on the scale of the work
(Bucci et al., 2021; Guzzetti et al., 2012; Santangelo et al., 2015).
Different types of inventories can be prepared. Event inventories (E-LIMs, Antonini et al., 2002; Ardizzone et al., 2012;
Donnini et al., 2017; Fiorucci et al., 2017; Harp and Jibson, 2017; Mondini et al., 2012; Santangelo et al., 2022) report
landslides that were triggered by specific events (i.e., rainfall, rapid snowmelt, volcanic eruption, earthquake).
Geomorphological inventories (G-LIMs) report landslides that can be recognised by geomorphologists usually from the expert
interpretation of stereoscopic aerial photographs (Bucci et al., 2021; Cardinali et al., 2001; Guzzetti et al., 2012; Santangelo et
al., 2015) but also LiDAR derived images are widely used (Niculiţă et al., 2016; Razak et al., 2013; Schulz, 2004; Van Den
Eeckhaut et al., 2007). They can be seen as the result of many landslide events over thousands of years (Malamud et al., 2004).
Multi-temporal landslide inventories (M-LIMs) include the information of G-LIMs and also report landslides that occurred in
the last tens of years as visible on historical images available at (possibly) regular time steps of several years (Galli et al., 2008;
Guzzetti et al., 2012; Zumpano et al., 2020).
G-LIMs can be prepared for wide areas and provide a fundamental source of information about landslides that occurred in the
last tens of thousands years. This particular type of inventory is an invaluable source of data but is time consuming and requires
a high level of training and experience to get accurate and reliable results (Guzzetti et al., 2012), which usually hampers the
systematic preparation of G-LIMs at regional or even national/continent scale.



This paper presents a new G-LIM prepared for the Daunia mountains in Puglia Region, Southern Italy. For this area,
historically affected by diffuse slow moving large landslides, the Regional government required the preparation of a new
landslide inventory to overcome the inhomogeneity among the existing inventories in terms of spatial distribution of landslides
and of techniques adopted and working and publication scales. The landslide inventory map was produced through the
interpretation of two sets of stereoscopic aerial photographs, taken in 1954/55 and 2003, supplemented by targeted field checks.
The new landslide inventory is an entirely original dataset openly accessible at
https://doi.pangaea.de/10.1594/PANGAEA.942427 (Cardinali et al., 2022).
**2 Study area**
The study area extends for 1,460 km$^2$, in the northern-western sector of the Apulia (Puglia in Italian) region (Southern Italy)
(**Figure 1**). The area corresponds to the Daunia Apennine, located in the external (i.e. eastern) part of the Southern Apennine
fold and thrust belt system.
The southern Apennine is the result of compressive tectonic dynamics, characterised by an oblique collision of the Calabrian
forearc with the Apulian margin (Filice and Seeber, 2019) evolved from an antecedent subductive phase of thrust-emplacement
(Vitale et al., 2011; Vitale and Ciarcia, 2013). Currently, the external (eastern) margin preserves the compressive tectonics
while the internal sector is characterised by an extensive dynamic with tension faults that dissect the pre-existent fold and
thrust structure (Brozzetti et al., 2009; Schiattarella et al., 2003).
The western sector of the Daunia Apennine is characterised by a wide variety of formations, mainly clay-rich flysch lithologies
having different mechanical properties, highly affected by folds and faults (Cotecchia et al., 2020; Losacco et al., 2021;
Pellicani et al., 2014b), resulting in medium-to-high relief with the elevations ranging between 55 and 1152 m a.s.l., where the
highest peak is represented by Monte Cornacchia. Morphologically, it is strongly influenced by both the prolonged tectonic
stresses and by the differences in composition and erodibility, where selective erosion has alternatively produced gentler and
rounded slopes on the more erodible clayey formations, and steep slopes cut on harder rocks. In this sector, the orography
associated with the widespread presence of clay-rich materials and the intense deformation is the main cause of landslides
(Ciarcia et al., 2003; Losacco et al., 2021; Spalluto et al., 2021; Wasowski et al., 2010; Zumpano et al., 2020). The eastern
sector is characterised by the Sub-Apennine clay formation and alluvial deposits, and by gentle slopes slightly dipping towards
NE. In this sector, several orders of terraced fluvial deposits overlay the sub-Apennine clay formation. Here, mass movements
are mainly concentrated along the scarps of the terraces.
The Daunia Apennine is influenced by a Mediterranean sub-humid climate with mild and often wet winters and usually hot
and dry summers. In general, inter-annual temperature variations are significant and the total yearly precipitation seldom
exceeds 1000 mm (Wasowski et al., 2012, 2010). In the area, torrential streams flow NE-SW, draining the sediments from the
Daunia Mountains, through the Tavoliere Plain towards the Adriatic Sea.
Landslides are widespread in the area, especially in the western sector, where landslides are a major cause of damage to urban
settlements, to the road network, and to the stability of buildings. These features determine a direct effect on the development
and the economy of the area (Pellicani et al., 2014b, a; Zumpano et al., 2020).

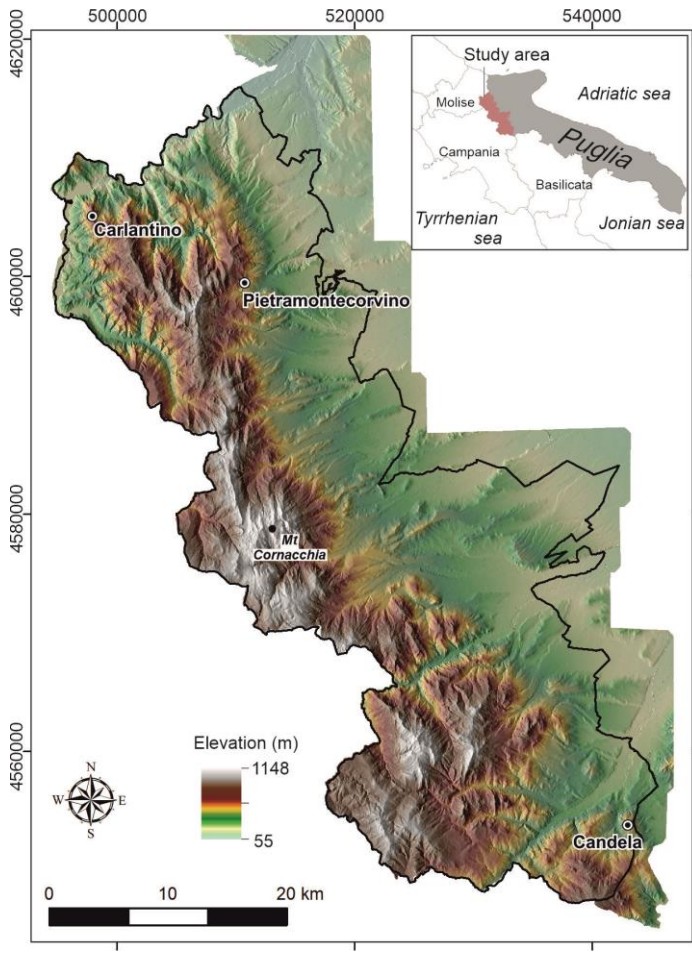


**Figure 1 -** Location map of the study area. Base map: DTM at 8 m resolution made available by the Regione Puglia
administration.



## 3 Materials and methods

The procedure adopted to prepare the landslide inventory map is illustrated in **Figure 2**. The overall procedure consists in three main phases: (i) data collection; (ii) photo-interpretation; and, (iii) editing. The entire work was carried out by a "mapping team" made up by four interpreters and one supervisor, and an "editing team" made by three GIS editors.

The first stage of the work consisted in the collection of data available for the study area (i.e., ancillary data, **Figure 2**; complete list in the Appendix A) and the pre-processing of the aerial-photographs to produce oriented stereo-models in absolute coordinates to be used for the visual interpretation in a photogrammetric GIS environment. Such photogrammetric pre-processing included the interior and exterior orientation of each pair of aerial photographs. For the interior orientation, a non-metric camera model was adopted for which the focal length of the camera, the flight altitude and the pixel size were required, while for the representation of the ground-to-image geometry (exterior orientation) Ground Control Points (GCPs) were used. The planimetric coordinates (x, y) of GCPs were manually chosen by visual comparison of the aerial photographs and an orthophoto available taken in 2006 (at 1 m resolution available for download at http://www.sit.puglia.it/) and 1988 (at 1 m resolution, available as WMS service at http://www.pcn.minambiente.it/mattm/servizio-wms/). The altimetric coordinate (z) of the GCPs were obtained from a Digital Elevation Model (8 m resolution available for download at http://www.sit.puglia.it/). Stereo models were prepared for both aerial photographs epochs, 1954/55 and 2003, i.e. the oldest and the latest aerial photographs acquisitions available for the entire study area.

The second phase of the work is the photo-interpretation (**Figure 2**) of the aerial photographs. At the very initial stage of the work, the entire mapping team defined a legend through an expeditious photo-interpretation in representative geomorphological settings and according to well-established landslide classifications schemas (Cruden and Varnes, 1996, Hungr et al., 2014, WP/WLI, 1993). Then, divided in pairs, for each stereo-pair the mapping team prepared a preliminary interpretation (1st level G-LIM in **Figure 2**) that was then reviewed by the supervisor and discussed with the mapping team. After this step, an updated version of the map was produced (2nd level G-LIM in **Figure 2**). Each of the two mapping team pairs performed their preliminary photo-interpretation in alternating strips along the flight plan. The side-lap between subsequent strips was therefore common between the different couples of interpreters, and it was used to compare, correct and homogenise the interpretation made by different sub-teams independently. Such a continuous interaction among geomorphologists made it possible to best define the characteristics of the identified landslides, limiting interpretation inconsistencies among operators. Finally, at significant advancement steps, extensive field checks were performed by the entire mapping team to check, validate and correct the inventory based on field evidence. After the field check stage, the interpretation phase was concluded and the landslide mapping considered quasi-definitive, i.e., occasional changes might be needed if inconsistencies were observed in the third stage (e.g., consistency with contour maps and hydrography).

The third and last step of the procedure, (editing) consists in the preparation of the geographical database. The interpreted features drawn as polylines by the mapping teams were verified on the digital topography, i.e., contour lines at scale 1: 5,000 (editing phase in **Figure 2**). The checked polylines were then converted into polygon features, and verified with topological





checks to avoid overlapping. Finally, the polygons were coded according to the adopted legend (legend description is available in Section 4 and the database schema is described in the metadata of the digital archive available at https://doi.pangaea.de/10.1594/PANGAEA.942427, Cardinali et al., 2022). Landslides recognised in the study area were classified by type according to Cruden and Varnes, (1996) and Hungr et al., (2014). Additionally, according to WP/WLI, (1993) landslides were classified based on the estimated depth (as shallow or deep-seated), and the inferred relative age (as relict, very old, old or recent). Examples and descriptions of landslides of different type, relative age, and depth, as well as of other geomorphological elements are detailed in Section 4. A more detailed explanation of the criteria used for the definition of the legend was given by Bucci et al., (2021).

## 3.1 Available data

The Geomorphological landslide inventory map (G-LIM) of Daunia Sub-Apennines was prepared through systematic visual interpretation of a set of 270 b/w stereoscopic aerial photographs taken in 1954/55 at 1: 33,000 scale and of 384 b/w stereoscopic aerial photographs taken in 2003 at 1: 30,000 scale. Aerial photographs were provided as 800 dpi scanned copies. All the ancillary data used for interpretation are listed and described in detail in Appendix A.

## 3.2 Hardware and software

For the interpretation phase we used the digital stereoscopic vision 3D PluraView System, composed of two monitors for the stereoscopic vision of digital images and a 45° inclined mirror placed on the bisector of the two monitors equipped with passive 3D glasses, (*https://www.3d-pluraview.com/en/*). A dedicated computer, equipped with IMAGINE Photogrammetry software (*https://www.hexagongeospatial.com/products/power-portfolio/imagine-photogrammetry*) was used for the pre-processing of digital stereoscopic images, and ArcGis software with the 3D Analyst extension tool, was used to digitise the 3D data in a GIS environment (Ardizzone et al., 2013; Fiorucci et al., 2015). The spatial resolution of the images (~1m for 1954/55 epoch, ~0.9m for 2003 images) and the zoom capability of the 3D PluraView System allow the identification of very small features. The stereo pairs were investigated using a zoom level at scale 1: 2500 to obtain mapped features compatible with a publication scale of 1: 5000. The capabilities of ERDAS IMAGINE ™ Photogrammetry Suite and ArcGIS ™ Stereo Analyst of preparing, managing and viewing multiple sets of oriented images all having the same reference system allowed the fast comparison of landscape features in different epochs, improving the interpretation.



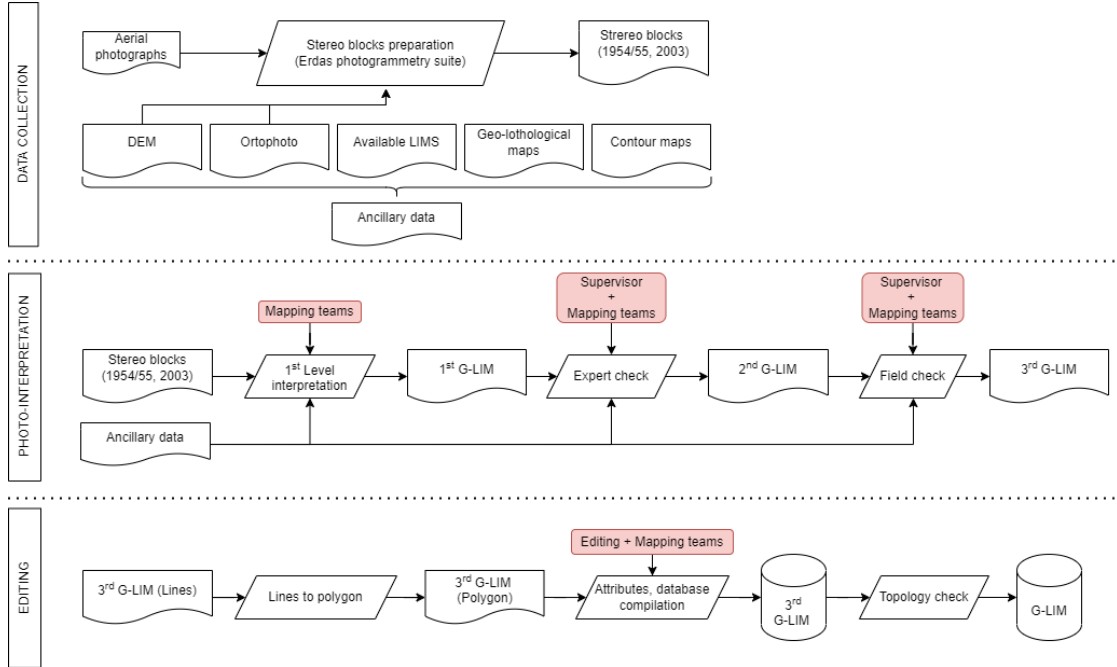

**Figure 2**. Flowchart of the procedure to prepare the geomorphological landslide inventory map. Pink rectangles indicate contributions of different teams/operators.

# 4 Geomorphological Landslide Inventory Map.

The inventory covers an area of 1,460 km$^2$ and include 17,347 landslides (**Figure 3**), corresponding to an average density of about 15.6 landslides per square kilometre (**Table 1**) if lowlands plains are excluded. Locally, landslide density reaches 60 landslides per square kilometre. Landslide size (Landslide Area, $A_L$ in m$^2$) is in the range $1.9 \times 10^1 < A_L < 6.8 \times 10^6$ (**Table 1**). Overall, landslides cover an area of 442 km$^2$, which represents the 39% of the hilly and mountainous portion of the study area.

**Table 1**. Descriptive statistics of landslides by relative age.

|  | Number of features | Total Area [m$^2$] | Minimum area [m$^2$] | Maximum area [m$^2$] |
|---|---|---|---|---|
| **Very old relict landslides** | 37 | $8.85 \times 10^7$ | $1.38 \times 10^5$ | $6.78 \times 10^6$ |
| **Very old landslides** | 120 | $8.75 \times 10^7$ | $5.82 \times 10^4$ | $2.86 \times 10^6$ |
| **Pre-2003 landslides** | 14,793 | $3.39 \times 10^8$ | $7.1 \times 10^1$ | $2.3 \times 10^6$ |
| **Recent landslides (2003)** | 2,049 | $5.01 \times 10^6$ | $1.89 \times 10^1$ | $1.63 \times 10^5$ |
| **Widespread landslides** | 348 | $1.13 \times 10^7$ | $3.78 \times 10^2$ | $3.54 \times 10^5$ |
| **Entire inventory** | 17,347 | $4.421.89 \times 10^8$ | $1.89 \times 10^1$ | $6.77 \times 10^6$ |

## 4.1 Landslide by relative age

Based on their appearance in aerial photographs, landslides were classified according to their relative age, namely: (i) very old relict landslides, (ii) very old landslides, (iii) old landslides, and (iv) recent landslides (**Figure 3**). The four relative age levels are the result of a major subdivision, referring to the general morphologic appearance of landslides (Keaton and DeGraff, 1996), and are based on the assumption that evidence of landslides become less obvious with the increasing age, due, for example, to erosion processes, vegetation growth, and occurrence of other landslides. Hence, older landslides are more difficult to detect than more recent ones.

In our G-LIM, the most represented landslides in the study area are old landslides (**Table 1, Figure 3C**), corresponding to 14,793 landslides covering a total landslide area of 339 km$^2$. Recent landslides (**Figure 3D**) are also very numerous (2,049) They are generally small, and cover a total area of 5 km$^2$. Very old relict and very old landslides (**Figure 3A, B**) are represented by 37 and 120 landslides respectively, and cover an area of 88 and 87 km$^2$. The feature of being fewer and larger than more recent landslides is common to other geomorphological inventories (Bucci et al., 2021, 2016a; Santangelo et al., 2015). The spatial relationships between landslides belonging to the relative age groups are displayed in **Figure 4**, where a representative example of the landslide relative age pattern for the entire inventory is shown.

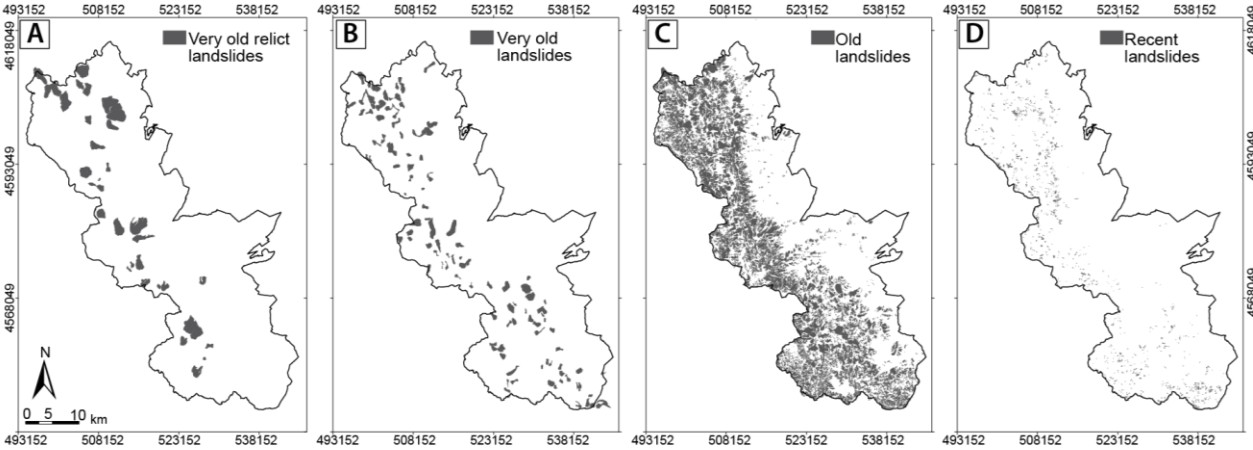

**Figure 3.** Spatial distribution of (A) very old relict landslides, (B) very old landslides, (C) old landslides and (D) recent landslides.

**Figure 4B-E** shows the inventory disaggregated according to the four relative age classes recognised. The sequence of the four panels represents a timeline, where landslides of each time step are shown with their own colour, the same as **Figure 4A**, whereas landslides belonging to antecedent time steps are represented in white.

In addition to the four mentioned relative age classes, a generation principle based on crosscut relationships, defines minor
subdivisions within each age class: younger landslides cover the older ones. For the overall inventory, this criterion allowed
us to detect up to: i) two landslides generations within the relict landslides, ii) two generations within the very old landslides,
iii) four generations within the old landslides, iv) two generations within the recent landslides. Such minor subdivisions are
applicable only to landslides that overlap, i.e. it is not applicable for landslides that do not overlap. For instance, in **Figure 4D**
the age class "old" records up to three overlapping classes, indicated respectively in light orange (first failure "old"), orange
(second failure "old") and red (third failure "old").

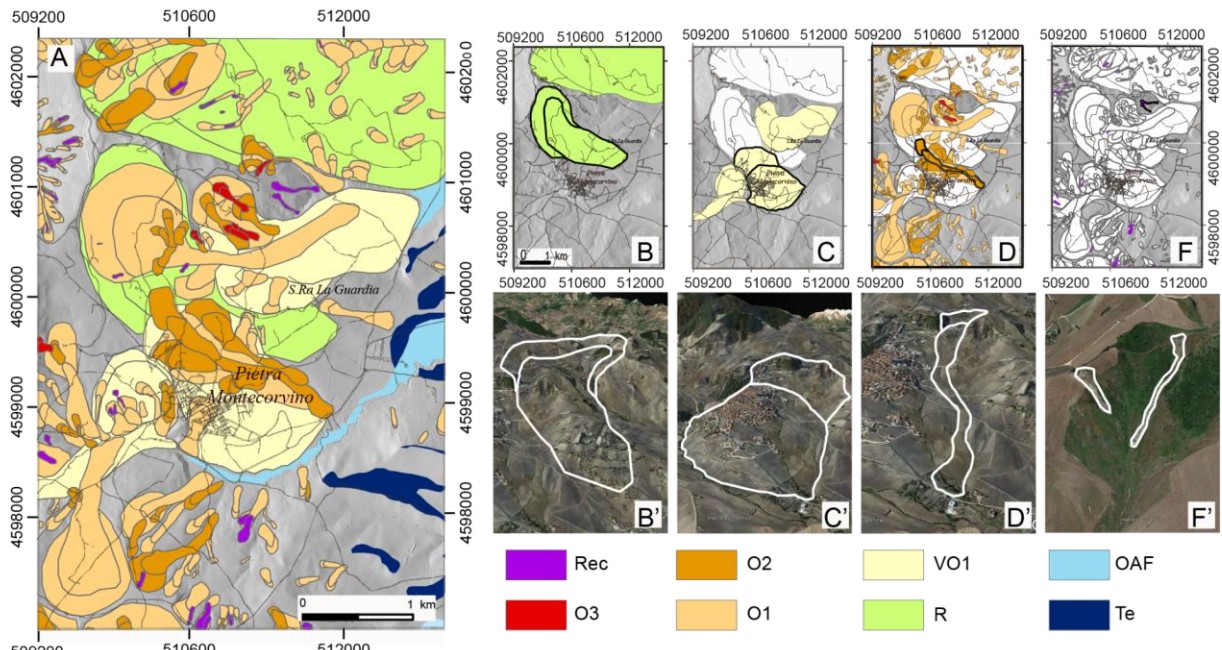

**Figure 4 -** Example of the relative age classes in the area of the Pietramontecorvino municipality (see **Figure 1** for reference). (A) Detail of the inventory in the selected location. Black outline polygon indicates the area in B to I. Very old relict landslides (B), very old landslides (C), old landslides (D), and recent landslides (E) are portrayed in detail. Black outlined polygons in B to E are represented singularly in a perspective view on © GoogleEarth™ in B' to F'. In the legend, Rec: Recent landslides; O1, O2, O3: old landslides of first, second and third generation; VO1: Very old landslides of first generation; R: relict landslides; OAF: old alluvial fan; Te: terrace.



Landslides classified as very old relict (**Figure 4B**) generally affect entire slopes, where the energy of the relief is greatest.
They are usually controlled by the geological and lithological structure and the presence of main faults (Bucci et al., 2016a).
Very old relict landslides are deeply dismantled by the erosive action of watercourses and often reshaped by recurring
gravitational phenomena. These morphological modifications are often related to the regional morphological and tectonic
evolution, which has determined considerable base level variations in the study area. As a consequence, these landslides are

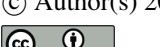



often suspended with respect to the present day base level, and totally or partially isolated from the recent evolution of the drainage network. Therefore, very old relict landslides are considered to have occurred under geomorphological and climatic conditions different from the present day (WP/WLI, 1993).

Landslides classified as very old (**Figure 4C**) show evidence of active and passive interaction with river dynamics, being often partly eroded from them and locally modifying their path. Also, morphological modifications of very old landslides can be induced by the occurrence of other mass movements over time. Very old landslides are generally large and in agreement with the present day river network, they are mainly distributed where the local relief is high and they often occur in or near very old relict landslides, as local reactivations.

Landslides classified as old (**Figure 4D**) present recognizable evidence on both 1954/55 and 2003 aerial photographs epochs. These landslides generally present morphological characteristics typical of landslides which are not modified by erosion (e.g., concave-convex shape of the slope, presence of steps and back slopes in the deposit area). Old landslides tend to cluster spatially, as it is evident from the several generations of mass movements that were recognised. Spatial clustering of old landslides is even more evident in the vicinity of very old or very old relict landslides. Old landslides are hypothesised to have occurred in more recent time (yet undefined) compared to very old and very old relict landslides.

Landslides classified as recent (**Figure 4E**) are represented by landslides that show evidence of having occurred close to the date of the 2003 aerial image. The diagnostic elements for the identification of recent landslides refer not only to the morphological evidence but above all to the spectral elements (photographic tone and contrast) which can be perceived even with a bi-dimensional (2D) view of the images. Recent landslides are mainly shallow failures essentially involving the soil horizons or the alteration of the debris cover for a few metres in depth. Their spatial distribution is ubiquitous and only limited to the areas where the triggering event induced the landslides. Therefore, they were found both in previously unaffected slopes or within pre-existing landslides.

As opposed to the recent landslides, which evidence is stronger, some uncertainty characterises the delineation of older landslide borders, which boundaries can be affected or even dismantled by different degrees of erosion, or covered by more recent slope failures. General considerations on relative age criteria of classification (Keaton and DeGraff, 1996) support our own experimental observations on landslide morphological appearance that the uncertainty degrees in landslide mapping increase with the increasing landslide age (Bucci et al., 2021). However, we do not have information on absolute age of the landslides in our study area, and only hypothesise that the majority of the mapped landslides occurred in the last 10–20 Kyr, as suggested by findings of recent studies on landslide age in Southern Apennine (Gioia et al., 2011) and elsewhere (Niculiță et al., 2016; Pánek et al., 2014).

**4.2 Landslide by type**

The type of the landslides is assigned based on the morphological and radiometric signature of each landslide on the aerial photographs. Landslide types included in the legend of the G-LIM are (i) slide type landslides (which include deep seated rock





slides or earth slides, shallow soil slides or earth slides), (ii) earth flow, (iii) slide-earth flow, (iv) debris flow, (v) rock fall and
(vi) sackung (**Table 2**).

**Table 2.** Descriptive statistics of landslides by type.

| Landslide type | Number of features | Total Area [m²] | Minimum area [m²] | Maximum area [m²] |
|---|---|---|---|---|
| **Slide** | 9,867 | $2.9 \times 10^8$ | $1.9 \times 10^1$ | $6.8 \times 10^6$ |
| **Earth flow** | 2,435 | $4.6 \times 10^7$ | $3.6 \times 10^1$ | $1.3 \times 10^7$ |
| **Slide-earth flow** | 4,657 | $1.6 \times 10^8$ | $8.8 \times 10^1$ | $2.9 \times 10^6$ |
| **Debris flow** | 38 | $6.5 \times 10^5$ | $1.4 \times 10^2$ | $1.4 \times 10^5$ |
| **Rock fall** | 1 | $6.0 \times 10^4$ | $6.0 \times 10^4$ | $6.0 \times 10^4$ |
| **Sackung** | 1 | $5.0 \times 10^5$ | $5.0 \times 10^5$ | $5.0 \times 10^5$ |
| **Total** | 16,999 | $4.3 \times 10^8$ | $1.9 \times 10^1$ | $1.3 \times 10^7$ |
| **Widespread landslide** | Number of features | Total Area [m²] | Minimum area [m²] | Maximum area [m²] |
| **Shallow earth/soil slides and earth flows** | 138 | $7.1 \times 10^6$ | $5.2 \times 10^2$ | $3.5 \times 10^5$ |
| **Debris flow** | 193 | $4.1 \times 10^6$ | $3.8 \times 10^2$ | $1.4 \times 10^5$ |
| **Rock fall** | 17 | $1.3 \times 10^5$ | $5.2 \times 10^2$ | $2.4 \times 10^4$ |
| **Total** | 348 | $1.1 \times 10^7$ | $3.8 \times 10^2$ | $3.5 \times 10^5$ |


According to the type of movement, the most represented landslides in the study area are slides (9,867 landslides, **Figure 5A**)
covering a total landslide area of 287 km². The 4,657 slide-earth flows (**Figure 5B**) cover an area of 158 km², whereas earth
flows are represented by 2435 landslides (**Figure 5C**) covering an area of 46 km². **Table 2** reveals that these three classes
represent more than 95% of the total landslides, while debris flows (38), rock falls (1) and sackungs (1) are rare.

The main features of the three more represented landslide types are illustrated in **Figure 6**. Inspection of the figure highlights
the main cartographic differences (**Figure 6A, C, E**) between different landslide types (**Figure 6B, D, F**).
**Figures 6A, B** illustrates the slide type landslides. Such failures present a well-defined scarp which can be semi-circular
(rotational slides) or angular (translational slides). The slide deposit is convex, with a morphologically depressed head of the
deposit, characterised by a centripetal drainage and local counter-slopes, and a toe characterised by a typical upward bulge
(**Figure 6B**). It is reasonable to expect a volume balancing between escarpment and deposit areas since slides are usually
characterised by low mobility of landslide material. This is clear in **Figure 6A, B** that shows a planimetric and altimetric shift
of about 25 metres of the displaced material, with no evidence of chaotic rearrangement and/or volumetric changes.
Slide-earth flows (**Figure 6C, D**) initiate as slides, then evolving into flows. Therefore, they show the characteristics of slides
(most commonly rotational) in the escarpment area and in the head of the deposit, where local endorheic conditions can
promote seasonal swamps with diffuse organic soils development (**Figure 6C, D**). On the other hand, the transport zone and



accumulation zone are more similar to earth flows, characterised by the typical lobe shape of the accumulation zone (**Figure**
**6C, D**).

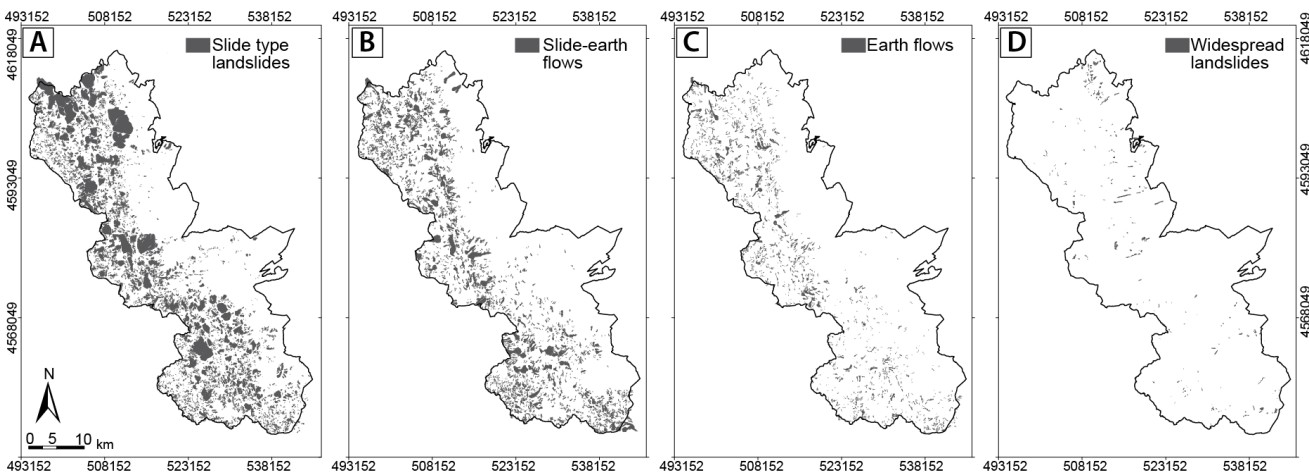

**Figure 5.** Spatial distribution of (A) slide type landslides, (B) slide-earth flows, (C) earth flows, (D) widespread landslides.


Earth flows (**Figure 6E, F**) are characterised by an overall elongated planar shape, with the median part narrower than
detachment and accumulation zones. The mobility of earth flows is generally higher than slides, which is also evident in their
more elongated shape. A typical earth flow is usually bounded by lateral streams draining the deposits into the main river
valley, which typically experiences narrowing of the valley section and erosion on the opposite side of the landslide toe (**Figure**
**6F**)
Debris flows and rock falls are not statistically represented in our inventory because the Daunia Apennine lacks the
environmental conditions favouring their development, such as steep rocky slopes and sub-vertical cliffs. However, local
geomorphological conditions (e.g. steep edges of suspended fluvial terraces) can promote the formation of coalescing
phenomena of small debris flows and small rock falls, which cannot be mapped individually but were included within areas
with widespread landslides (**Figure 5D**, Section 4.4). Finally, sackungs are poorly represented in our study area.

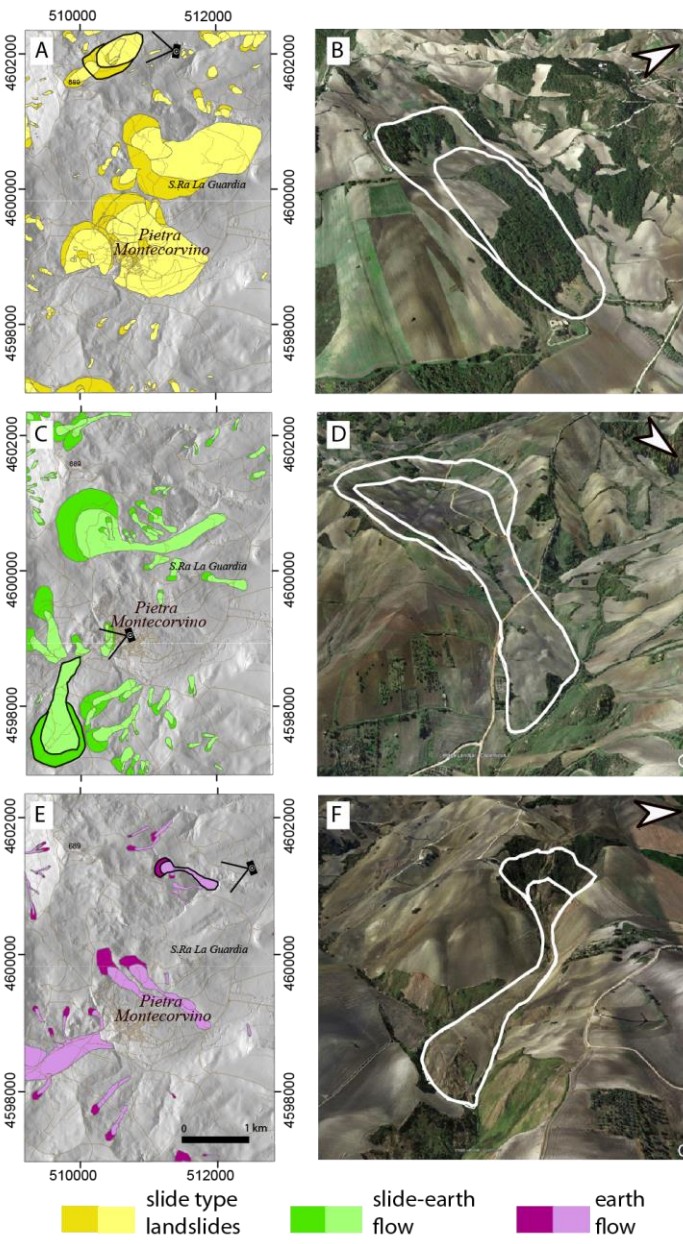

**Figure 6.** Examples in map (left column) and on © GoogleEarth™. images (right column) of slide type failures (A,B), slide-earthflow landslides (C,D), earthflows (E,F). For each row, black outlined polygons and camera symbols in the left image identify the landslide shown in the corresponding image on the right and the point of view, respectively.



**4.3 Landslide estimated depth**

The estimated depth of a landslide is assigned based on the main morphological characteristics of the landslide, such as: slope height, and extent of curvatures along the slope (convexity and concavity). Based on these morphological characteristics, landslides were classified as deep-seated or shallow.

Deep-seated landslides are predominantly represented by failures that cover considerable extents, even exceeding one square kilometre. Many of the deep-seated landslides can be classified as very old relict, very old and old. They often involve considerable volumes of material, and can alter the local morphology and geological structure. In the inventory map, deep-seated landslides are represented with two polygons, distinguishing the deposit area from the scarp area.

Shallow landslides are generally small ($\sim 10^4$ m$^2$) and mainly represented by slide type and flow type landslides. They are generally characterised by a scarp not very pronounced, with an estimated height of less than 2 metres, and by a deposit without evident concavities and convexities on its surface. In the inventory, shallow landslides are mapped as single polygons including both the scarp and the deposit area.

**4.4 Widespread landslides**

The areas of widespread landslides (**Table 2**, **Figure 5D**) are represented by landslides which size is smaller than the smallest feature that can be represented at the publication scale of the final map. In the map we distinguished between: i) shallow earth/soil slides and earth flows, ii) debris flows, iii) rock falls. Their spatial distribution is mainly related to locally steep slopes, for instance in the vicinity of badlands, in the crown areas of pre-existing large landslides, or along the edges of ancient suspended fluvial terraces (**Figure 7**). Widespread shallow slides and flows typically involve weathered clayey soils, while widespread debris flows and rock falls are commonly fed by loose debris or poorly cemented conglomerates and breccias. Finally, it was not possible to assign a well-defined age to widespread landslides, although their morphological evidence suggests seasonal reactivations.

**4.5 Descriptive statistics of the G-LIM**

Descriptive statistics of landslide number and size are shown in **Figure 7**. Box plots in **Figure 7A, B** show the distribution of landslide areas for the landslide classified based on the expected relative age (**Figure 7A**) and the type of movement (**Figure 7B**).

Inspection of the plot in **Figure 7A** confirms that slides, slide-earth flows and flows are the most represented landslides of the inventory. Among these three classes, the slide type landslides are characterised by the largest size variability, while the slide earth flows are on average slightly larger than the others. In the plot of **Figure 7A**, landslides (on the left) are separated by areas of widespread landslides (on the right) because the latter are not related to a single feature and cannot be directly compared to single landslides. This is the reason why the areas of widespread landslides are on average larger than the single



302 landslides. In particular, the areas of widespread soil slides and earth flows are larger than the others because they typically

303 involve edges of fluvial terraces characterised by strong lateral continuity.

304 The plot in **Figure 7B** shows the size distribution of the four relative age classes of landslides. The clear separation of the age

305 classes according to their median values is a statistically robust indication of the reliability of this landslide classification

306 strategy, which is underused in the landslide-oriented international literature.

307 Comparing **Figure 7A** and **7B**, it is worth noting that the median area of all landslide types is around $10^5$ m$^2$, which corresponds

308 to the median area of the old landslides group. This was expected since this is the most represented landslide age class within

309 the inventory.

310 **Figure 7C** shows landslides count (represented by a colour gradient and labels) and cumulated area (proportional to circle

311 sizes), grouped by relative age classes and landslide type. Visual inspection of the plot reveals that relict and very old landslides

312 are not represented by small landslides, and in particular by debris flows and rock falls, which tend to be small and easily

313 obliterated by erosion and subsequent failures. Further evidence is that a large portion of the total landslide area is occupied

314 by few relict and very old kilometre-scale slope failures, whereas a decreasing trend both in cumulated area and total number

315 of landslides is evident through time for all landslide types within the old landslides age class. Such evidence suggests a size

316 threshold effect of previous landslides on subsequent slope movements, probably controlled by slope-scale morphological and

317 hydrological perturbations induced by the occurrence of the first failure.



**Figure 7**. Plots summarising landslide statistics. (A) Landslides number (represented by a colour gradient and labels) and cumulated area (proportional to circle sizes), grouped by relative age classes and landslide type (SK, sackung; SEF, slide earth flow; S, slide; RF, rock fall; DF, debris flow; EF, earth flow). (B) Distribution of landslide areas within each relative age class. The relative age classes are indicated through a letter (R, relict; V, very old; and O, old), and a subscript number that indicates the generation within the age class (1–4). (C) Scatterplot showing cumulative size and count of landslides grouped by relative age in the abscissa and type in the ordinate. Landslide number is represented by a colour gradient and labels; total area is proportional to circle sizes.


### 4.6 Geomorphological elements


In the landslide inventory map, some geomorphological elements are also reported in addition to the mass movements (**Figure**
**8**). Such elements can be considered in relation with slope evolution and can provide useful information for landslide
identification and mapping. In particular, our map includes alluvial deposits, fluvial terraces, and alluvial fans. Alluvial
deposits are flat and always located in the lowest portions of the *Tavoliere delle Puglie* Plain and along the main water courses
draining the hilly and mountain areas of the Daunia Apennine. Older alluvial/fluvial deposits are nowadays suspended over
the present-day base level and are organised in fluvial terraces at different elevations, recording the ongoing deepening of the
drainage network during the Quaternary. Alluvial fans were recognized and mapped in two relative age classes according to
their appearance. They were classified as relict if dismantled and dissected due to the incision of the present-day river network.
They were instead classified as recent if well preserved.

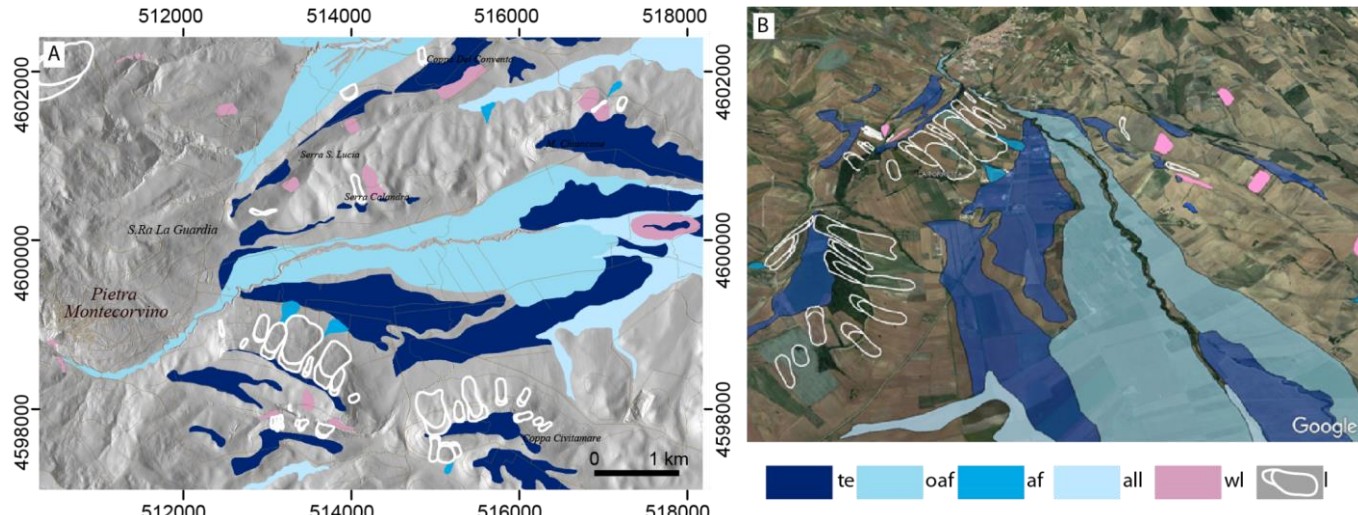

**Figure 8** - (A) Excerpt of the landslide inventory map. Landslides were coloured in white for enhancing the geomorphological elements. (B) Birds Eye view of the same area as (A) prepared using © GoogleEarth™.


### 5 Conclusion


The landslide inventory map presented in this paper is a new catalogue of landslide phenomena in the Daunia Apennine. It
provides new data uniformly distributed over the study area at an unprecedented cartographic detail.
The new landslide inventory map was produced upon request of the authorities of the Puglia Region, to solve the heterogeneity
and limitations in the currently available landslide inventories (Pellicani et al., 2014b; Pellicani and Spilotro, 2015), mainly
due to: i) partial coverage of the territory, ii) heterogeneity in the degree of cartographic detail, iii) lack of metadata and other
methodological information, iv) geospatial inconsistencies where the different products overlap.



The new inventory represents a significant improvement upon earlier mapping because: (i) it presents a new homogeneous landslide coverage for the Daunia Apennine, (ii) it is acquired with a uniform method and based on rigorous and reproducible criteria, (iii) it keeps the same cartographic detail throughout the entire area, and (iv) it presents a systematic classification of landslides by type, relative age, and depth, hence providing basic information at 1: 5000 scale for landslide characterization and related land management activities. In addition, the spatially distributed information on landslide coverage portrayed in the map, and the detailed scale of mapping, provide the fundamental information for landslide susceptibility and hazard assessment (Guzzetti et al., 2005), and for supporting and integrating the study of seismic microzonation and the assessing of seismic hazard both in urban and rural areas (Vignaroli et al., 2019). In addition to the technical and land management benefits, our inventory provides new data to further investigation of the Quaternary evolution of the landscape of this part of the Southern Apennines.

Analysis of the data revealed that the great majority of landslide volume was mobilised by relict and very old landslides, whereas the landslide maximum size decreased over time and tended to cluster around pre-existing failures. The results have relevance to determine the statistics of landslide size (Malamud et al., 2004), and to trigger detailed studies on the types, patterns, and distribution of landslides in relation to geology (Bucci et al., 2016b) structure (Bucci et al., 2013), tectonic (Bucci et al., 2014) and climatic forcing (Schiattarella et al., 2017). The presented dataset also documents the relationships between landslides and other geomorphological elements, in particular with ancient and recent alluvial deposits and alluvial fans. These fluvial related landforms can be used as input data to study quaternary depositional (Mirabella et al., 2018) and erosional events (Mancini et al., 2020), and in combination with landslide information, to study the interplay of gravitational and fluvial processes (Santangelo et al., 2013).

We acknowledge that scarcity of data represents a limitation and a challenge for the landslide community. Most regional-scale studies are less effective (or analysed) than expected due to the lack of easy sharing of landslide information in digital format. Our work is also intended as a contribution to the broader geomorphological community, promoting the sharing of landslide dataset at regional scale from various environments.

Finally, it is worth reminding that a landslide map, even if accurate, does not answer all questions regarding hazards or risk at any scale. A Landslide map is informative about the area covered by landslides, but nothing can be said about the remaining territory. The map dataset should be used as an indication to optimise resources and to plan investigations aimed at determining landslide hazard and risk at a larger scale. In our case, the map dataset is published at 1: 5000 scale, and should be consulted at most at the same detail level.





## 6 Authors contributions

We declare that all authors' contributions in preparing both the dataset and the manuscript are equivalent to that of a first author. This is why all authors appear in alphabetical order.

## 7 Acknowledgments

This work was supported by the Civil Protection of the Puglia region, in the framework of the project "*Integrated assessment of geo-hydrological instability phenomena in the Apulia region, interpretative models and definition of rainfall thresholds for landslide triggering*" funded by the P.O.R. Puglia 2014-2020, Asse V - Azione 5.1. (Project identification number: B82F16003840006).

## 8 Data availability

Dataset is available at: https://doi.pangaea.de/10.1594/PANGAEA.942427 (Cardinali et al., 2022).

## Competing interests

The authors declare that they have no conflict of interest.

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

## Appendix A

In the first phase of work, information and cartographic/thematic products were collected to support the preparation of the
geomorphological landslide inventory map of the Daunia Apennines. Some of the listed products have been provided by the
Regione Puglia, others can be consulted on the web, through Cartographic Portals or Web servers:
●   Regional Technical Maps in digital format at a scale of 1:5,000, supplied by Regione Puglia (www.sit.puglia.it);
●   Geological maps at scale 1:50.000 (CARG), available on the ISPRA cartographic portal;
●   Geological maps at a scale of 1:100,000 (Geological Map of Italy), available on the ISPRA cartographic portal;
●   Hydrogeomorphological map in scale 1:25.000 available on the SIT site of the Regione Puglia
(http://webapps.sit.puglia.it/freewebapps/Idrogeomorfologia/index.html);

●   Digital Elevation Model (DEM, cell 8×8m) provided by Regione Puglia (www.sit.puglia.it);
●   Digital Elevation Model (DEM) made by LiDAR (1×1m cell), carried out within the framework of the Extraordinary
Plan for Environmental Remote Sensing (PST-A) and provided by the Ministry of Ecological Transition (MiTE).

●   Urbanised and land use maps, provided by the Regione Puglia (www.sit.puglia.it);
●   Orthophoto maps provided by the Regione Puglia (www.sit.puglia.it);





● Landslide inventory maps of Apulia from:
■ *IFFI Inventory*: Inventory of Phenomena Franosi in Italy, compiled by ISPRA;
■ **Official Archives**: *Piano Stralcio per l'Assetto Idrogeologico* (PAI) of the Basin Authorities of Puglia,
Basilicata, of the Rivers Trigno, Biferno and minor Saccione and Fortore, and of the Rivers Liri-Garigliano
and Volturno; *Project Inventory of Phenomena Franosi in Italy (IFFI - update 2006)*, compiled by ISPRA;
*Census Project of Italian Areas Historically Vulnerated by Geological and Hydraulic Disasters (AVI)*, drawn
up by the National Group for Defence against geo-hydrological Disasters of the National Research Council
(GNDCI-CNR);
■ **AdBP photo interpretation**: landslides from the photo interpretation study conducted by AdBP (Basin
Authority of Puglia Region);
■ **Province of Foggia**: it deals with the landslides surveyed by the Province of Foggia within the framework
of the agreements signed with the AdBP concerning the "Activity of collection and classification of data on
landslide phenomena in the Province of Foggia" and "Conduct of studies for the deepening of the aspects
related to the classified landslide phenomena".