# Peer review of "Geomorphological landslide inventory map of the Daunia Apennines,"

_Earth System Science Data, 2022_

## Referee Comment (RC2)

[referee-annotated manuscript omitted]

---

## Author Comment (AC1)

This document contains a point-to-point answer to the comments of the two reviewers. Reviewer comments are in italics, indented. Responses are in plain text, not indented. Quotations from the manuscript and text that will be added in the revised version of the manuscript are in italics between inverted commas, not indented.

**Reviewer #1**

*This paper outlines the method and initial results of a landslide inventory map for the Daunia Apennines region. Landslides are presented in terms of age and type, with reference to spatial occurrence and some statistics. I would like to congratulate the authors on producing such a well-written paper. The writing is excellent. It is engaging, well-formed and very well referenced. It flows well and is easy to read from start to finish. In addition it does not get bogged down in technical language in the methods while still communicating the complexity of the research project. Figure 2 nicely illustrates the development process of the LIM. As such, I do not have any minor comments, only some more broad thinking points to be addressed.*

We thank the reviewer for her very positive comment.

*In the early manuscript there is a slight lack of clarity. My main confusion was what exactly they were inventorying. Is the G-LIM pertaining to soil or rock landslides?*

About the type of material involved, we did not apply any a-priori selection, which is why we did not mention it. On the other hand, we specified the material involved in the classification scheme used (See Table 2 and description in the text). We did find, though, that in Table 2 we wrote "*slides*", whereas that should at least read "*slide type*" to include any type of material, as already done at lines 236-237. This will be modified accordingly in the revised manuscript.

*Are they events which have already occurred completely, or slow moving slides in progress? This information is needed early on to engage readers.*

About the concept of landslides mapped in the G-LIM being events that occurred "completely" in the past or slowly moving landslides in progress, in the Introduction we stated: "*Geomorphological inventories (G-LIMs) report landslides that can be recognised by geomorphologists usually from the expert interpretation of stereoscopic aerial photographs ... but also LiDAR derived images are widely used ... They* [G-LIMs] *can be seen as the result of many landslide events over thousands of years.*". A few lines on we also wrote: "*G-LIMs can be prepared for wide areas and provide a fundamental source of information about landslides that occurred in the last tens of thousands years*". We meant by this that we included in our inventory all landslides that could be recognised, regardless of being or not stabilized at the moment the aerial photographs were taken, as no one would have that information from aerial photographs. On the other hand, heuristic analysis of such images allows to derive information about landslides relative age, which we did include in our dataset.

*More than 50% of the abstract is introduction material. There is space to achieve more depth about the specifics of the study and move this information to the introduction.*

About the information in the abstract, we are not certain to get the comment right (i.e. which is the 50% referred to). We maintain that the first half of the abstract should contain text that defines the general scientific relevance of the topic and the research problem/question. The second half should specify without too many details the scientific answer (method and some results and their relevance). This is why we would like to keep the abstract as it is, without adding too many details that are already in the text. About moving the general part of the abstract in the introduction (if this is the 50% of the abstract the reviewer is referring to), we think it is already there, even though not all in one place. Precisely: (i) lines 11-14 are elaborated at lines 24-31; (ii) lines 14-17 are elaborated at lines 58-61.

*The temporal factor needs to be clarified. Since the main data sets bracket a period of time (1954-2003 and 1988-2006), was the G-LIM updated to resolve geomorphic changes during that period? Was the development of a slide taken into account in the delineation? More concrete detail would be welcomed in the first pages of the ms.*

We thank the reviewer for this comment. First of all, the period 1988-2006 is not to be considered as those orthophotos were not used to map landslides (they are not stereoscopic) but only for the external orientation of aerial photographs in the digital stereoscope (lines 111-114). About the second part of the comment, in our work we have defined different age levels: relict and very old landslides did not show modifications in the time interval 1954-2003. Landslides classified as pre-2003 and 2003 consider by definition the latest evidence in this 49 years interval. To better explain this point we will add text (after line 176) that reads: "*Landslides modifications within the two epochs (1954 and 2003) were negligible for very old relict and very old landslides, whereas they were more frequent for old landslides (i.e. pre-2003). Therefore, the final landslide delineation considers by definition the latest evidence within the 49-years time interval.*"

*There is some conceptual information missing around what is considered a landslide in this study. These specific types of landslides need to be put into context to global landslide understanding. Firstly, it seems like the object mapped include both a potentially active portion, and the deposit of a failure. Secondly, there are mostly flows and slides of soil(?) material, or the highly weathered carapace of weak rock (if I connect the dots between the geological setting and the results correctly). I understand these are typical landslides types in Italy, but since the journal is international the context is important. This context is also important since the end-use of this LIM, as mentioned in the text, should inform planning decisions by e.g. the regional authorities.*

We are not sure to get the point the referee is making here. As commonly understood, a landslide is the movement of a portion of rock/soil/debris along a slope under the effect of gravity. Hence, a landslide map is the representation of the geomorphological evidence of landslide phenomena occurred in an area, according to the basic assumptions stated in the Introduction section (lines 32-41). Furthermore, we did not include any activity information as aerial images do not provide such evidence. If detectable and compatible with the representation scale, landslide polygons are split in two areas: the source area (landslide scarp) and the deposit area, as specified also in Section 4.3. That being said, the scarp and deposit polygons bring no information about landslide activity. Furthermore, as the reviewer correctly guesses, the relative abundance of landslide types is directly linked to the morpho-lithological setting, as always is. But we do not really understand what she is asking by pointing out this aspect referring to a "global landslide understanding". Earth flows or soil slides are not only common in Italy. We acknowledge that we have only shortly commented, about landslide size/type/age/depth distribution in the dedicated sections since the journal specifications require little to no analyses to be carried out. So, if the reviewer is

asking to comment about the relationship between the morpho-litho-structural setting/context and landslide abundance and types, we maintain it is out of the scope of this presentation of the dataset and will be extensively studied and included in a specific outcoming paper, as already stated at lines 350-352. Anyways, we considered adding a short comment in the description of the study area to partially address this comment and a different comment of Reviewer #2. At line 94 the text will read: "*In response to the litho-structural and morpho-climatic setting of this area, landslides are widespread. They are mostly slow moving slide-type and flow-type movements, involving soil and earth, and secondarily rock material* (Wasowski et al., 2012, 2010; di Lernia et al., 2022; Zumpano et al., 2020). *Rapid moving (i.e. debris flows) and fast moving (i.e. rock falls) landslides are less abundant. The widespread presence of landslides in the area is a major cause of damage…*"

*Can the authors make any inferences about the meaning of the data, in terms of why they see what they see? For example, how is there only one rockfall across an entire mountain range? How do the results connect back to the study setting, and what is the implication of the results in terms of the end-use of this LIM going forward?*

As stated in the previous answer, we have decided not to carry out in-depth analyses about landslide distribution and the morpho-litho-structural setting, as suggested by the journal guidelines. On the other hand, we do get what the referee is suggesting. That is why at lines 266-270 the text already reads: "*Debris flows and rock falls are not statistically represented in our inventory because the Daunia Apennine lacks the environmental conditions favoring their development, such as steep rocky slopes and sub-vertical cliffs. However, local geomorphological conditions (e.g. steep edges of suspended fluvial terraces) can promote the formation of coalescing phenomena of small debris flows and small rock falls, which cannot be mapped individually but were included within areas with widespread landslides (Figure 5D, Section 4.4)*". In terms of use of this map from different end-users, we considered adding this sentence in the Conclusions: "*In the G-LIM presented, the vast majority of landslides are slow-moving, whereas rapid and fast landslides are rare (debris flows and rock falls). Such evidence should be further confirmed by in-depth studies (e.g. multi-temporal inventories or susceptibility models) as it may indicate a relatively low exposure of human life compared to other mountain areas of Southern Apennines. On the other hand, it must be taken into account that the increasing frequency of extreme rainfall events may trigger rapid and fast moving landslides also in this area, differently from what was observed in the past. This aspect further shows that systematic landslide mapping to prepare new inventories and update the existing ones is crucial, particularly in the ongoing climate change scenario (Donnini et al., under review).*"

*Figure 1: you might consider adding a broader location map to indicate location within Europe for those not familiar with southern Europe.*

Figure 1 will be amended as suggested.

*Table 2 and section 4.2: A description of what is meant by 'widespread landslide' types is missing. Now I found it in section 4.4, but I am still confused about what it means. More information is needed, or an explanatory figure/image.*

We understand that finding a definition only in section 4.4 may be misleading. So we anticipated a sentence right before Table 2 where we state: "*In Table 2, widespread landslides refer to polygons representing groups of landslides whose size is smaller than the smallest feature that can be represented at the publication scale of the final map.*"

*The information about other geomorphic elements comes as a surprise. This should be mentioned earlier on or otherwise appears as an attempt to shoehorn extra data into the study. Is the method for this part of the study mentioned?*

For how the paper is structured, we acknowledge that this section comes in the end, probably unexpected. So we added a short paragraph in the beginning of Section 4 that introduces all the elements that will be analytically described in specific sub-sections. The text reads: "*In this section the different groups of elements composing the inventory will be analytically described. Sections 4.1 to 4.3 describe landslides classified according to their relative age, type classification and estimated depth. Section 4.4 describes the widespread landslides (i.e. landslides represented as groups of failures due to their small size and high spatial frequency). Section 4.5 describes the descriptive statistics of the landslide inventory, and Section 4.6 presents the geomorphological elements (i.e. elements that are considered in relation with slope evolution and that can provide useful information for landslide identification and mapping)*". We did not consider adding any further information about the method as it is the same used for landslide mapping.

**Reviewer #2**

*The paper is very well written, very clear and concise, and important for the scientific community, as it deals with the very basis of any susceptibility/hazard/risk analysis, i.e. landslide inventory. The entire conceptual framework is clearly explained, and the data and methodology are properly described. The results are representing a good projection of the previous chapters. The graphic part is conclusive and properly supports the text. There would be some minor comments to address, inserted in the attached document.*

We thank the reviewer for this very positive comment and appreciation of our work and presentation.

*L28: Maybe a newer citation would be better, since this one (undoubtedly, a key one) is already 10 years old.*

Agreed. We added a newer reference. In the new version of the manuscript the references cited will be: "*(Guzzetti et al., 2012; Bucci et al., 2021)*"

*L95: A brief/general description of landslides typology (closely linked with 4.2.) would be necessary.*

According to this suggestion we will add the following text at line 94 to briefly characterize landslide types in the area: "*In response to the litho-structural and morpho-climatic setting of this area, landslides are widespread. They are mostly slow moving slide-type and flow-type movements, involving soil and earth, and secondarily rock material. Rapid moving (i.e. debris flows) and fast moving (i.e. rock falls) landslides are less abundant. The widespread presence of landslides in the area is a major cause of damage…*"

*L172: I suggest to keep only "relict", as it already implies "very old".*

Whereas it is true that any relict landslide is a very old failure, it is not true that any very old landslide is relict. It is not only a matter of age but of the spatial relationship of the landslides

with the present-day morphology. A detailed description of relict landslides is provided in Section 4.1, and the substantial difference between very old and very old relict landslides is stated at L204-207: "*...these* [relict] *landslides are often suspended with respect to the present day base level, and totally or partially isolated from the recent evolution of the drainage network. Therefore, very old relict landslides are considered to have occurred under geomorphological and climatic conditions different from the present day (WP/WLI, 1993)*".

*L237: Would this* [slide-earth flow] *be the same with "compound landslide"? Personally, I have rarely encountered this expression.*

Slide-earth flow is mentioned by Hungr et al., (2014) among the complex failures.

*L240: Type of material missing, to be in accordance with the rest.*

Correct. In Table 2, "Slide" is missing material here because they can involve different materials (rock, earth, soil). To be concise and consistent with other parts of the paper we will modify it in "*slide-type*" to include any possible material as already stated at L236-237.

*L329: The short forms should be inserted in the descriptive text above, to allow the reader an easier correlation.*

Correct. The figure caption will be amended as suggested. It will read: "***Figure 8*** *- (A) Excerpt of the landslide inventory map. Landslides were colored in white for enhancing the geomorphological elements. (B) Birds Eye view of the same area as (A) prepared using the ©GoogleEarth ™. In the legend: te, fluvial terrace; oaf, old alluvial fan; af, alluvial fan; all, alluvial deposits; wl, widespread landslides; l, landslide.*"

The following references will also be added to the reference list:

*di Lernia, A., Cotecchia, F., Elia, G., Tagarelli, V., Santaloia, F., and Palladino, G.: Assessing the influence of the hydraulic boundary conditions on clay slope stability: The Fontana Monte case study, Engineering Geology, 297, 106509, https://doi.org/10.1016/j.enggeo.2021.106509, 2022.*

*Donnini, M., Santangelo, M., Gariano, S.L., Bucci, F., Peruccacci, S., Alvioli M., Althuwaynee, O., Ardizzone, F., Bianchi, C., Bornaetxea, T., Brunetti, M.T., Cardinali, M., Esposito, G., Grita, S., Marchesini, I., Melillo, M., Salvasti, P., Yazdani, M., Fiorucci, F.: Landslides triggered by an extraordinary rainfall event in Central Italy on September 15, 2022, under review.*